# Congestion-Free Ant Traffic: Jam Absorption Mechanism in Multiple Platoons

**Prafull Kasture \*** and **Hidekazu Nishimura**

Graduate School of System Design and Management, Keio University, Yokohama, Kanagawa 223-8526, Japan
\* Correspondence: kp_prafulla@a3.keio.jp

**Abstract:** In this paper, an agent-based model of ant traffic on a unidirectional single-lane ant trail is presented to provide better understanding of the jam-free traffic of an ant colony. On a trail, the average velocity of ants remains approximately constant irrespective of density, thereby avoiding jamming. Assuming chemotaxis, we analyze platoon-related scenarios to assess the marching-platoon hypothesis, which claims that ants on a trail form a single platoon in which they march synchronously, thereby reducing hindrances due to increasing density. Contrary to that hypothesis, our findings show that ants on a trail do not march synchronously and do experience stop-and-go motion. However, more interestingly, our study also indicates that the ants' chemotaxis behavior leads to a peculiar jam absorption mechanism, which helps to maintain free flow on a trail and avoids jamming. Again, contrary to the marching-platoon hypothesis, our findings also indicate that, rather than assisting traffic flow, forming a single cluster actually triggers jamming.

**Keywords:** ant-trail; self-organization; swarming; jam-free traffic; intelligent transportation system

## 1. Introduction

Recent advancement in computation has made it possible to record, simulate, and analyze multi-agent complex systems by using agent-based modeling [1,2]. Agent-based modeling is widely used for the modeling and synthesis of complex distributed systems such as distributed financial markets, artificial intelligence, and town planning [2–4]. Agent-based modeling is also used to study spatiotemporal organization in systems of interacting agents, including vehicular traffic, molecular motors moving on polymeric tracks, and ant colonies moving on trails [5–8]. One of the major uses of agent-based modeling is the analysis of emergent behaviors in interacting multi-agent systems, which cannot be understood by merely studying individual elements of a complex system. In recent years, the study of ants' emergent behavior on trails has attracted particular attention for several reasons. The vast trail systems that ants form for transportation share many of the features of vehicular transportation systems [9–17]. Individual trails created by an ant colony can be functional for hours and can be considered analogous to highways. Thus, the collective movement of ants on trails ("ant traffic" (AT)) is analogous to vehicular traffic on a highway network [5,18]. The social behavior of ants also indicates that biological evolution may have optimized AT; examples include (i) the formation of three lanes in bidirectional AT and (ii) natural selection of the shortest path to a food source [10,16,17,19–23].

A recent empirical study of unidirectional AT in *Leptogenys processionalis* revealed several exciting properties [14]. The average velocity of ants on a trail remained almost independent of density on that trail. Consequently, no jamming phase (referred to simply as "jamming") was observed in the fundamental diagrams. It was also found that ants on a trail formed clusters. The fundamental diagrams for AT are in contrast to those of vehicular traffic. In the latter, the average velocity decreases at high density, and thus, jamming occurs, indicating stop-and-go behavior and congestion [14].

Inspired by these fascinating findings about AT, a few previous theoretical studies attempted to explain the mechanism for jam-free AT on a unidirectional trail [9,13,15]. In [13], it was assumed that ants on the trail detect their headway distance visually and control their propelling force according to the headways. The model in [13] was able to capture all the features observed in [14], and it also indicated that ants might have nearest-neighbor repulsion, which was previously unknown. However, previous physiological studies about ants indicated that ant species with trail pheromones are practically blind and cannot detect objects farther than a few millimeters [10,21]. On the other hand, in the case of [15], it was assumed that ants use chemotaxis for traffic management on AT. Specifically, the work in [15] argued that ants reduce their velocity for a corresponding increase in pheromone concentration. The model in [15] also captured all the major observations from [14], and it also supported the findings of [13], which suggested that ants might have nearest-neighbor repulsion. However, although the above-mentioned assumption of [15] partially agreed with previous studies (ants use chemotaxis for traffic management), previous studies showed that ants are attracted towards a higher concentration of pheromones (in contrast with the assumption in [15], ants might be increasing their velocity when they discover a higher concentration of pheromones) [10,21,24,25].

Meanwhile, the model in [9] can be considered a pioneer model in the traffic flow analysis of AT, which has been well analyzed [5,6,11,12,26,27]. The model in [9] is simple, yet it is in accordance with previous studies. However, it is a general model of ant traffic, which was proposed and analyzed prior to [14]. Therefore, it should be further improved and analyzed in light of the findings from [14]. Further improvement and analysis of the model in [9] might give us a better understanding of jam-free AT from [14].

In the present paper, we propose an improvement to the model from [9], where, similar to [9], we use asymmetric hopping and particle exclusion on a discrete lattice to represent single-lane unidirectional motion; whereas, as an improvement over the previous model (where the pheromone concentration had only two discrete levels (presence or absence)), pheromone concentration levels in our model are continuous. Another difference between the two models is that the ants in [9] traveled with constant distance in one unit of time, whereas the probability of motion depends on the pheromone (stochastic dynamics). On the other hand, in our model, we assume a constant probability of motion, but a distance to be traveled in one time step is continuous and depends on the pheromone concentration (deterministic dynamics). All the assumptions and improvements in the new model are based on previous physiological studies, and the model (in certain limits) also captures all the main features of the experimental results from [14], which verifies that the model in certain limits is able to mimic real-life ants. Using the model, we also analyze different platoon scenarios to understand platoon-related phenomena in AT. These platoon-related analyses test the marching-platoon hypothesis. The marching-platoon hypothesis has two components: (1) platoon formation, whereby ants on a trail converge to form platoons, and (2) marching ants, whereby ants inside a platoon march together (move with the same instantaneous velocity). In the marching-platoon hypothesis, as the ants move with the same instantaneous velocity (synchronized march), they (ants) do not experience the hindrance that usually arises in traffic systems with increasing density [14]. Thereby, ants on a trail maintain a constant average velocity, regardless of their density (in [13], the marching platoon was referred to as the infinite cluster, where it was assumed that agents inside the platoon have a low level of noise, which ensures that all agents have almost the same propulsion force (synchronized march)). We also investigate the inter-platoon dynamics in AT. Our analyses give us significant new insight into the mechanism behind the jam-free AT found in [14].

## 2. Model and Simulation Scenario

### 2.1. Model

Ants on a trail communicate with each other by chemotaxis: as they move forward on the trail, the ants drop and sense chemical substances known as pheromones [10,21–23]. The ants'

movements depend on the local pheromone concentration ahead of them, which they detect with their antennae [10,21]. Physiological studies of ants and other insects show that the antennae-detected pheromone concentration is converted into a self-propelling force [10,21–25]. This force increases with the pheromone concentration until the concentration ($\sigma_{sat}$) that saturates the antennae [10,21,24,25]. The antennae cannot detect any further increase in concentration above $\sigma_{sat}$, and the propulsion force for concentrations beyond saturation remains approximately that at saturation [10,24,25]. Based on the aforementioned chemotaxis behavior of ants, we present an agent-based model of the behavior of ants on a unidirectional single-lane ant trail (the ant-trail model (ATM)). As explained earlier, our model is an improvement on the model by [9].

The model in [9] was analyzed using cellular automaton. However, considering that the observations of individual ants are an integral part of our analysis, we chose agent-based modeling. For the simulations, we used NetLogo as the modeling platform, which is a multi-agent programmable modeling environment [28]. The model in NetLogo comprises two types of agent, namely (i) stationary agents representing the cells of the trail (environment in the model) and (ii) moving agents representing the ants. Each cell of our one-dimensional ant trail can accommodate, at most, one ant at any time step (see Figure 1). The cells are labeled by the index $i$ ($i = 1, 2, \ldots, L$), where $L$ is the length of the trail. We associate the following three numerical variables with each cell.

- The binary variable $s_i(t)$ is either zero or one depending on whether the cell is empty (zero) or occupied (one) by an ant at time step $t$.
- The pheromone concentration $\sigma_i(t)$ is a numerical variable ranging from zero to $\sigma_{\text{sat}}$. $\sigma_i(t) = 0$ means that there is no pheromone in cell $i$ at time step $t$, whereas $\sigma_i(t) = \sigma_{\text{sat}}$ means that the cell is saturated with pheromone at that time step.
- $\Omega_i$ represents a resistance or drag that the cells present against the ant motion. $\Omega_i$ represents factors related to trail conditions such as obstacle, rough trail, or uphill, which have a negative impact on the motion of ants. In short, $\Omega_i$ provides opposition to the motion in the preferred direction, which is represented in ATM by a negative velocity (velocity in the opposite direction). In a given simulation, $\Omega_i$ is constant for a given cell. To avoid ants moving backwards, $\Omega_i$ is designed to always be positive, but less than the non-zero minimum ant velocity ($v_{min}$) (as explained later, $\Omega_i$ is used while defining the heterogeneous trail.).

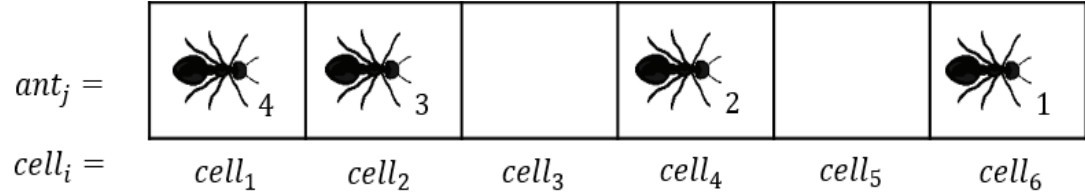

**Figure 1.** Schematic representation of the ant-trail model (ATM), showing a single-lane unidirectional ant trail from left to right. Each cell is indexed by $i$, and each ant is indexed by $j$. At any given time, each cell can contain only one ant.

As shown in Figure 1, the ants are also labeled with a unique number $j$ ($j = 0, 1, 2, \ldots, N$), where $N$ is the total number of ants in the simulation at the time of measurement (as explained later, the number of ants changes over time). Each ant has the following two associated variables:

- $v_j(t)$ is the instantaneous velocity of ant $j$ at time step $t$. $v_j(t)$ is continuous and ranges from zero to one.
- $p_j(t)$ is the position of ant $j$ on the trail at time step $t$ and ranges from zero to $L$. Similar to $v_j(t)$, $p_j(t)$ is also continuous.

All the parameters from the model are summarized in Table 1. For the ATM simulations in the paper, we assumed that: (i) ants do not move backwards; (ii) the probability of forward motion is

constant; and (iii) the distance traveled in one time step depends on the pheromone concentration in the cell ahead. At each time step, the states of the model variables are updated in two stages.

**Table 1.** An overview of the parameters in the ant-trail model (ATM).

| Description | Symbol |
| --- | --- |
| Unique identity of a cell in the trail | $i$ |
| Presence or absence of an ant in the trail $cell_i$ at time $t$ | $s_i(t)$ |
| Pheromone concentration in the trail $cell_i$ at time $t$ | $\sigma_i(t)$ |
| Resistance by the trail $cell_i$ to the motion of an ant | $\Omega_i$ |
| Pheromone concentration saturation level | $\sigma_{sat}$ |
| Unique identity of an ant in the simulation | $j$ |
| Velocity of the $ant_j$ at time $t$ | $v_j(t)$ |
| Position of the $ant_j$ at time $t$ | $p_j(t)$ |
| Minimum velocity of an ant towards the cell with no pheromone and no other ant | $v_{min}$ |
| Trail length | $L$ |
| Evaporation rate | $er$ |

### 2.1.1. Stage I: Ant Motion

The first stage of the update represents the behavior of ants, which depends on the interaction of a given ant with its surroundings (surroundings of an ant include trail and other ants in the simulation). In this stage, based on the information about pheromones and the presence of ants in the cell ahead, the value of the instantaneous ant velocity $v_j(t)$ is generated for a given time step $t$. At the end of Stage I, we obtain the value of $p_j(t+1)$ for each ant and the scan value of $s_i(t+1)$ for each cell at time $t+1$. The ant positions and parameters are updated according to the following rules. If ant $j$ is in cell $i$, the instantaneous velocity of that ant from cell $i$ towards cell $i+1$ depends on $s_{i+1}(t)$, $\sigma_{i+1}(t)$, and $\Omega_{i+1}$, as follows:

$$v_j(t) = \begin{cases} 0, & \text{if } s_{i+1}(t) = 1 \\ \begin{cases} \max(v_j(t-1) - 0.1, v_{\min}) - \Omega_{i+1} & \text{with probability P} \\ v_j(t-1) - \Omega_{i+1} & \text{with probability (P-1)} \end{cases} & \text{if } s_{i+1}(t) = 0 \text{ and } \sigma_{i+1}(t) < 1 \\ (v_{\min} + \mathbf{a} \times \sigma_{i+1}(t) - \Omega_{i+1}), & \text{if } s_{i+1}(t) = 0 \text{ and } 1 \leq \sigma_{i+1}(t) < \sigma_{\text{sat}} \\ (v_{\min} + \mathbf{a} \times \sigma_{\text{sat}} - \Omega_{i+1}), & \text{if } s_{i+1}(t) = 0 \text{ and } \sigma_{i+1}(t) \geq \sigma_{\text{sat}}. \end{cases} \tag{1}$$

$$p_j(t+1) = p_j(t) + v_j(t). \tag{2}$$

For the time step given in Equation (1), the following four cases are possible.

The first case represents the action of an ant in cell $i$ if the next cell is occupied by another ant. In that case, the former ant cannot move forward, which is represented by $v_j(t) = 0$.

In the second case, the next cell contains no ant ($s_{i+1}(t) = 0$), but also no pheromone. Ants are sensitive to the pheromone concentration; they can detect a pheromone even at an extremely low concentration. For the simulations in this paper, we assumed that ants cannot detect pheromones below $\sigma_{i+1}(t) < 1 (<< \sigma sat)$. When no pheromones are present, the ant will not have the trail; thus, to avoid wastage of energy due to high velocity, we assumed that the ants reduce their velocity or move with $v_{min}$, whichever is higher, and the probability $P$ gives the probability of this event (change in velocity). Conversely, the agents maintain the same velocity as $t$ with the probability $(1 - P)$. In the second case, we assumed that $P = 0.7 (> 0.5)$, which represents high sensitivity of the ant to absence of the pheromone. After a velocity reduction, the final velocity is calculated with consideration of $\Omega_{i+1}$. For the simulations in this paper, $v_{min}$ was chosen in such a way that it would avoid backwards motion due to $\Omega_{i+1}$. Therefore, we needed to choose an $v_{min}$ value that was always greater than $\Omega$. On the other hand, a larger value of $v_{min}$ led to a smaller velocity range. Therefore, to maximize the velocity range, we chose $v_{min} = 0.15$ (for further explanation on $v_{min}$, check Appendix A.1).

In the third case, the next cell contains no ants ($s_{i+1}(t) = 0$), but it contains a detectable level of pheromones below saturation ($1 < \sigma_{i+1}(t) < \sigma_{(sat)}$). In that case, the instantaneous velocity of the ant depends on the pheromone concentration in the next cell. For the analysis presented in this paper, we use a deterministic equation to represent velocity changes related to the third case. Deterministic dynamics represents high sensitivity to pheromones, as well as low inertia of ants on the ant trail. In this scenario (the third case), the value of the prefactor **a** is decided based on $\sigma_{sat}$ and $v_{max}$ (upper velocity limit ($= 1$)). For the functioning of the model within given velocity limits, ($\mathbf{a} \times \sigma_{sat} \leq v_{max} + \Omega_{i+1} - v_{min}$) should be satisfied. On the other hand, a lower value of $\mathbf{a} \times \sigma_{sat}$ will lead to a lower velocity range. Therefore, considering the above restriction and mathematical simplicity, we selected $\mathbf{a} \times \sigma_{sat} = 0.8$, and assuming $\sigma_{sat} = 80$, we chose $\mathbf{a} = 0.01$ (for further explanation about $\sigma_{sat}$, check Appendix A.2).

In the fourth and final case, the next cell contains no ant, but the level of pheromone is above saturation. In that case, the velocity of the given ant becomes equal to the velocity at $\sigma_{sat}$, which can be calculated by using the third case equation ($v_{min} + \mathbf{a} \times \sigma_{sat} - \Omega_{i+1}$). As explained above, for the analysis in this paper, we assumed $\sigma_{sat} = 80$.

As specified by Equation (2), the new position $p_j(t + 1)$ of the given ant is calculated by adding the position of the ant at time $t$ to the distance traveled in unit time ($v_j(t)$). It is interesting to note that the dynamics presented in Equation (1) resemble the dynamics in the Nagel–Schreckenberg model where the velocity of an agent depends on the headway distance, whereas in the case of our model, the velocity of an agent depends on the pheromone concentration. This resemblance indicates that both models are analogous to each other, which indicates that there might be some similarities between the two traffic systems [29].

### 2.1.2. Stage II: Pheromone Updating

The second stage of the update represents changes in the environment (trail) over time due to the interactions of agents with it. At each time step, the pheromone concentration on the trail changes for two reasons: (i) evaporation due to environmental factors (normally, the evaporation rate ($er$) remains constant if the surroundings remain approximately unchanged) and (ii) pheromone accumulation due to further discharge of the pheromone by the ants (in one time step, an ant can release an amount $\tau$ of pheromone, referred to as a pheromone unit). At the end of Stage II, we obtain the subset $\sigma_i(t + 1)$ using the subsets $S_i(t + 1)$ and $\sigma_i(t)$, and the velocity of the ant in the cell as follows:

- Evaporation:

$$\sigma'_i(t + 1) = \sigma_i(t) - (\sigma_i(t) \times er), \quad if \quad \sigma_i(t) > 0. \tag{3}$$

- Accumulation:

$$\sigma_i(t + 1) = \begin{cases} (\sigma'_i(t + 1) + \tau), & if \quad s_i(t) = 1 \quad and \quad \sigma'_i(t + 1) < \sigma_{sat} \quad and \quad v_j(t) > 0 \\ \sigma_{sat}, & if \quad s_i(t) = 1 \quad and \quad \sigma'_i(t + 1) > \sigma_{sat}. \end{cases} \tag{4}$$

Herein, as we discussed, $\sigma_{sat} = 80$. As given by Equation (3) for evaporation, a certain fraction of the pheromone concentration evaporates from each cell at each time step depending on $er$. After evaporation, the remaining pheromone concentration on the cell is further affected by the addition of pheromone emitted by the ant in the cell at the same time step. As shown in Equation (4), for accumulation, there are two possibilities.

In the first case, a cell is occupied by an ant ($s_i(t) = 1$) that moved forward in the previous time step ($v_j(t) > 0$). Moreover, the pheromone concentration in the cell after evaporation ($\sigma'_i(t + 1)$) is below saturation. In that case, the ant releases a unit volume of pheromone that is added to the current pheromone concentration in the given cell.

In the second case, a cell is occupied by an ant ($s_i(t) = 1$), but $\sigma_i'(t+1)$ is above saturation. As such, there is no need to add any further pheromone because it would go undetected. In this case, the concentration remains at the saturation level $\sigma_{sat}$.

### 2.2. Simulation Scenarios

ATM has multiple variables (such as $er, \sigma_{sat}, \tau$), and in future studies, it will be interesting to investigate how different variables affect the system dynamics, but the purpose of this paper is to validate the model and test the marching platoon hypothesis. Therefore, herein, we define the following simulation scenarios.

### 2.2.1. Periodic Boundary Conditions and Introduction of New Ants

In this study, we wanted to analyze the AT on a trail that has formed over a period of time without any external interference. Thus, data collected for the analysis were always collected after a considerable time from the beginning of the simulation (data were collected from an established traffic flow). Furthermore, we applied horizontal periodic boundary conditions, making the simulation scenario equivalent to a circular trail by connecting the last cell ($cell_{1000}$) to the first ($cell_1$). Natural ATs are open boundary systems. However, as we wanted AT simulations to form over the period of time without any external interference, we used the periodic boundary condition. Nevertheless, at 1000 cells, the trail was long enough to make any ant self-interaction effects negligible (which arise under the periodic boundary condition with a short track).

At the beginning of the simulations, there was only one ant ($ant_0$) on the trail, and all cells of the trail were assigned no pheromone ($\sigma_i(0) = 0$). During the simulation, at each time step, if $cell_1$ was empty, we introduced a new ant there ($cell_1$) with a probability known as the inflow rate. At the time of introduction, the new ant was assigned the variable $j$ based on the value of Nbefore the introduction of the new agent, where for the new ant, the value of $j$ was equal to the above-mentioned N. With the addition of new ants, the density in the simulation increased with time. In the present study, we conducted simulations until the density reached its limit (density = 1), where no further addition of ants was possible. Although foraging ants use multiple recruiting mechanisms, new foragers are usually recruited slowly [21]. Therefore, we were able to assume that the inflow of ants onto the trail was sufficiently low. For the simulations presented herein, we used an inflow rate of $0.001 (<< 1)$, which allowed sufficient time for the traffic flow to become established in each density scenario.

### 2.2.2. Trail Scenarios

The simulations presented herein were conducted for the following two trail scenarios.

### Homogeneous Trail

In this scenario, we assumed that the trail was uniform, as represented by the cell variable $\Omega_i$. As mentioned above, $\Omega_i$ is a cell variable that represents the resistance of $cell_i$ against the forward motion of ants on the trail. On a homogeneous trail, all cells have the same value of $\Omega_i$, meaning that all cells offer the same resistance to each ant.

### Heterogeneous Trail

Usually, ant trails form on natural surfaces, meaning that different parts of the same trail might have differing structures and offer differing levels of resistance to the ant motion. This differing resistance can lead to AT bottlenecks. Therefore, to analyze an ant-trail system on a heterogeneous surface, we defined a two-part trail of which one part comprised low-resistance cells ($\Omega_i = 0$) and the other part comprised high-resistance cells ($\Omega_i = 0.1$). In this scenario, the low-to-high junction marked the beginning of the bottleneck, and the high-to-low junction marked the end of the bottleneck.

## 3. Fundamental Diagrams and Evaporation Rate

From a systemic point of view, the fundamental diagrams are "the most efficient way" to represent and analyze the flow characteristics of a traffic model [5]. The fundamental diagrams of the relationships between (a) the average velocity-density and (b) the flow-density, as shown in Figure 2, were obtained from ATM simulations with different *er* values on a homogeneous trail. For a time step in our simulation, we calculated (i) the average velocity ($v_{avg}$) by averaging the observed displacement of all the ants for that time step, (ii) the density (*d*) by dividing *N* at that time step with $L$ ($d = N/L$), and (iii) the flow (*f*) by multiplying *d* by $v_{avg}$ at the same time step ($f = d \times v_{avg}$). The fundamental diagrams in Figure 2 have the following characteristics: (1) different *er* values lead to different fundamental diagrams; (2) at low *er* values ($0 < er < 0.1$), the $v_{avg}$ values in the simulations have a non-monotonic relation with *d*, which leads to anomalous fundamental diagrams; (3) the fundamental diagrams for $er = 0$ and $er = 1$ have similar behaviors to that of the total asymmetric simple exclusion process (TASEP) model, but with different hopping probabilities [5]. All of the above-mentioned characteristics of our model are similar to those in the model by [9], which indicates that despite the improvements, the model presented in this paper has similar behavior to that by [9]. Based on the fundamental diagrams in Figure 2, the *er* value in ATM can be divided into three ranges as follows.

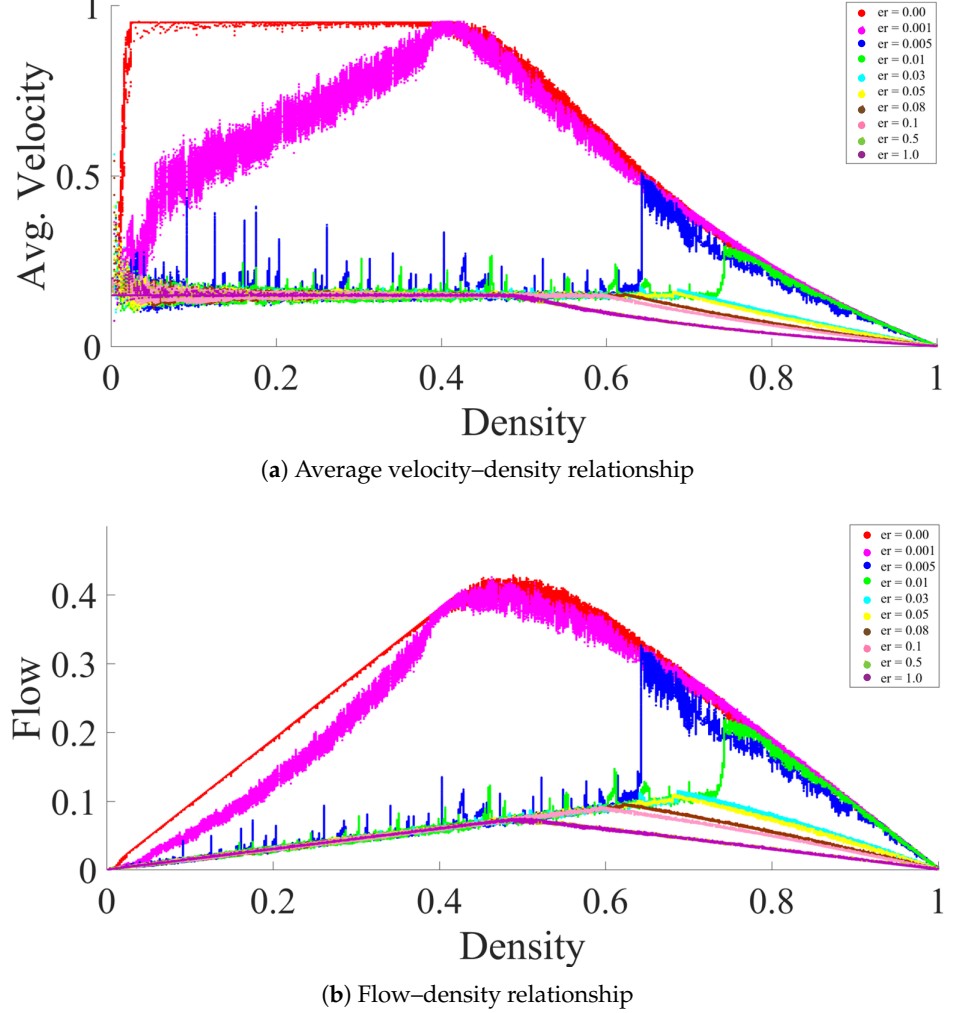

(**a**) Average velocity–density relationship

(**b**) Flow–density relationship

**Figure 2.** A fundamental diagram of the ATM simulation for different *er* values is plotted: (**a**) average velocity–density relationship (**b**) flow–density relationship. Parameters other than *er* were kept constant: $L = 1000$ cells, $\sigma_{sat} = 80$, $v_{min} = 0.15$.

*3.1. High to Medium Evaporation Rate (0.1 < er ≤ 1)*

As shown in Figure 2, for $0.1 < er \leq 1$, ATM behaves similar to the TASEP model with a low hopping probability. In this range, the pheromone on a trail evaporates quickly, leading to the trail with no pheromone. On a trail with high or medium *er*, most of the ants will not find any pheromone for most of the time, leading to an AT where pheromones play minimal roles in the traffic system. In such a case, most of the ants in ATM travel with $v_{min}$ (constant), which is similar to the TASEP model with a low hopping probability.

*3.2. Meager Evaporation Rate (0 ≤ er < 0.001)*

In this range, the fundamental diagrams of ATM are similar to the TASEP model with a high hopping probability. With the meager *er*, the pheromone on a trail almost never evaporates as long as there are ants on the trail, and the pheromone on the trail increases sharply, even with a small rise in *d*. In the ATM with a meager *er* value (apart from initial small *d*), the $v_j(t)$ of all ants remains approximately constant at a high value (close to the saturation level) or increases with *d*, because the ATM behaves similarly to the TASEP model, but with a high hopping probability.

*3.3. Low Evaporation Rate (0.01 < er < 0.1)*

In this range, the fundamental diagrams of ATM are interesting, where the $v_{avg}$ of ants in ATM remains constant for a wide range of *d*, leading to a linear rise in *f* (see Figure 2). In this range of *er*, the pheromone on a trail does not evaporate quickly. However, it also does not reach a saturation level. We call the trail mentioned above (a trail with low *er*) an active trail, where $\sigma_i(t)$ in a cell depends on the flow in the recent past. We assume that ants use this information (information about the flow in the recent past) to manage the velocity on the trail efficiently. Therefore, for further simulations in the paper, we used the *er* from the above range (*er* = 0.02).

**4. Model Validation**

For model validation, we compared data from the ATM simulations with the empirical data from [14], beginning with fundamental diagrams. The fundamental diagrams of the relationships between (a) the average velocity-density and (b) the flow-density, as shown in Figure 3, were obtained from ATM simulations for *er* = 0.02 (low *er*). Along with the *er* value, all parameter values used hereafter in the simulations are summarized in Table 2. As shown in Figure 3a, unlike a vehicular traffic system, for ATM, the $v_{avg}$ remained constant for a wide range of *d* values (*d* = 0 to ≈ 0.75). Consequently, the *f* on the trail increased linearly with *d*. Thus, in the fundamental diagrams for the present simulation, the free-flow phase was observed up to *d* ≈ 0.75. These observations (i.e., a free-flow phase and constant $v_{avg}$ up to *d* ≈ 0.75) are in basic agreement with the empirical results from [14], which validates the fundamental diagrams of our model. On the other hand, the same fundamental diagrams indicated that after specific density (*d* ≈ 0.75), due to the exclusion dynamics of the model, ants in the model experienced jamming, which was not observed in [14]. It is important to note that the data in [14] for *d* mentioned above (*d* > 0.75) were limited and, therefore, cannot be used for validation (in the future, we should make an effort to validate the exclusion assumption, as well as the jamming phase observed in ATM by experimentation). Another interesting thing to note is that $v_{avg}$ in Figure 3 is entirely independent of *d* in the free-flow phase, whereas in the case of empirical data from [14], the data showed a slight decrease in $v_{avg}$ with an increasing *d*. The models in [13,15] replicated the above-mentioned decrease in $v_{avg}$. In both models, agents' velocity decreased slightly at high densities (inside the platoon). In [13], the decrease was due to a progressive reduction of the headway, while in [15], it was due to a progressive increase of the pheromone concentration. In other words, both models suggested that there might be some weak force that slows down ants at high *d* values. Whether it is weak near-neighbor repulsion (as suggested in [13]) or a decrease in velocity due to the above saturation pheromone (as suggested in [15]) should be investigated in the future.

However, as the $v_{avg}$ decrease is minimal, for mathematical simplicity, the model in this paper did not consider the weak forces mentioned above.

**Table 2.** An overview of the parameter values from the ATM simulations.

| Description | Simulation Values |
|---|---|
| Pheromone concentration saturation level | $\sigma_{sat} = 80$ |
| Resistance from the trail $cell_i$ to the motion of an agent in the homogeneous trail scenario | $\Omega_i = 0$ |
| Resistance from the trail $cell_i$ in the high resistance section to the motion of an ant in the heterogeneous trail scenario | $\Omega_i = 0.1$ |
| Minimum velocity of an ant towards the cell with no pheromone and no other ant | $v_{min} = 0.15$ |
| Trail length | $L = 1000$ |
| Inflow rate | $= 0.001$ |
| Evaporation rate | $er = 0.02$ |
| High resistance trail section in the heterogeneous trail scenario | $cell_{400}-cell_{800}$ |

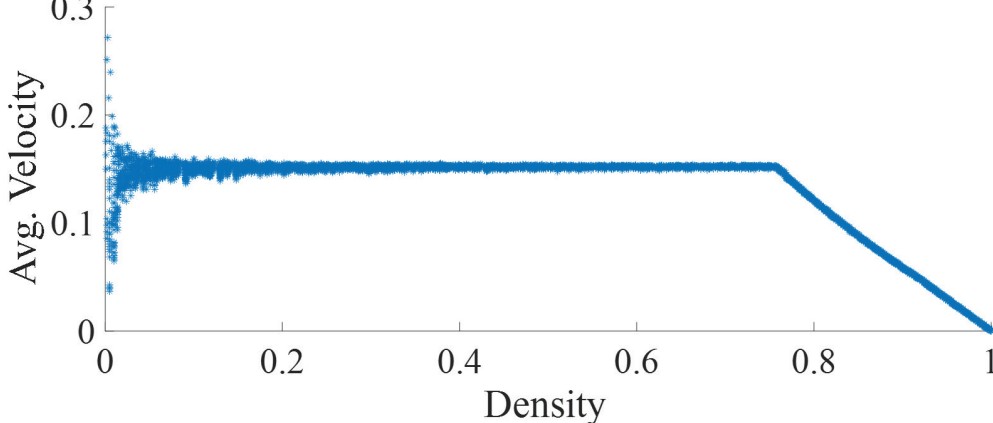

(**a**) Average velocity–density relationship

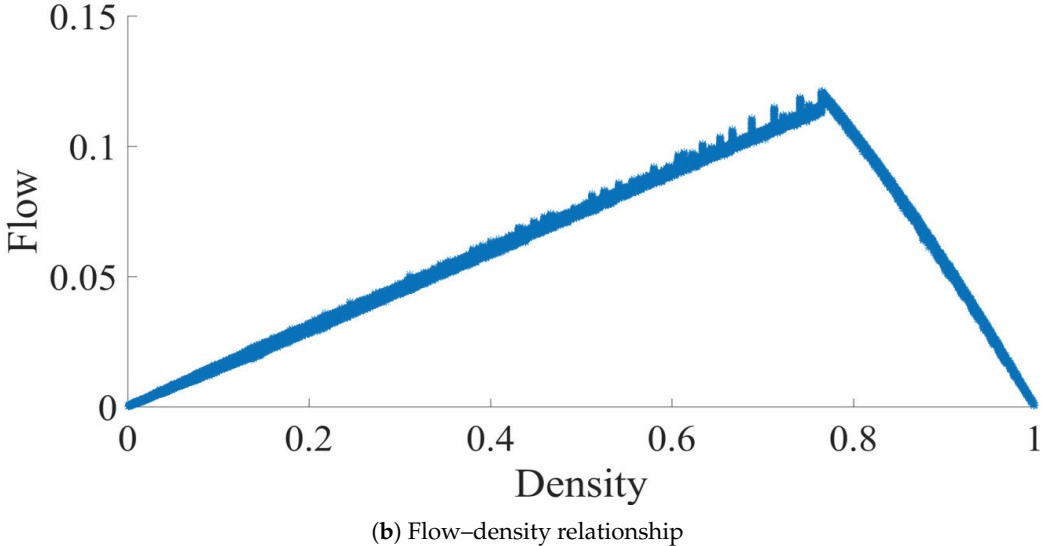

(**b**) Flow–density relationship

**Figure 3.** (**a**) Average velocity and (**b**) flow of agents plotted against their density. Simulation scenario: $er = 0.02$, $L = 1000$ cells, $\sigma_{sat} = 80$, and $v_{min} = 0.15$.

Furthermore, as shown in Figure 4, we analyzed the distance headway distribution in the simulation, where at lower *d* values, large distance headways were predominantly found, which indicates a random-headway distribution. The distribution at lower *d* values was well described by a negative exponential distribution. On the other hand, at medium to higher *d* values [0.2, 0.7], the distribution became much sharper with an increasing *d* value, indicating platooning. For these *d* values, the log-normal distribution appeared to provide the best fit. The above observations related to distance headway and platoon formation were similar to the empirical observations in [14]. On the other hand, we also observed that there were no or very few ants who traveled with intermediate headways, especially for medium *d* values [0.2, 0.4]. This can be explained by platooning dynamics, where ants with intermediate or smaller headway accelerate due to the presence of pheromones until they catch up with preceding ants (platooning), whereas ants with a headway greater than an intermediate value do not experience pheromones, thus maintaining their velocity and avoiding joining a platoon ahead. This observation about the intermediate headway indicated that not all ants were converging to form a platoon.

We also compared the velocity distribution of ants in ATM with data from [14]. As shown in Figure 5, similar to the empirical data from [14], the distribution became much sharper with an increasing global density, whereas the most probable velocity remained approximately constant. This happened because, as explained earlier, at higher *d* values, most of the ants traveled in platoons, where the velocity of an ant was governed by the velocity of the ants ahead, as well as the pheromone concentration in the next cell. At higher *d* values (inside the platoon), the maximum velocity of an ant was limited by the velocity of the ant ahead. At the same time, due to a high concentration of pheromones (in a platoon, due to the ants ahead, the given ant experienced a larger pheromone concentration), the $v_j(t)$ of the given ant increased, which led to the scenario where most of the ants in the platoon traveled with a $v_j(t)$ higher than $v_{min}$ (as most of the ants experienced the pheromone concentration, no ants had deceleration due to a lack of pheromones, which led to a decrease in the overall number of ants who travel with $v_{min}$). The above two phenomena resulted in a decrease in the velocity fluctuation.

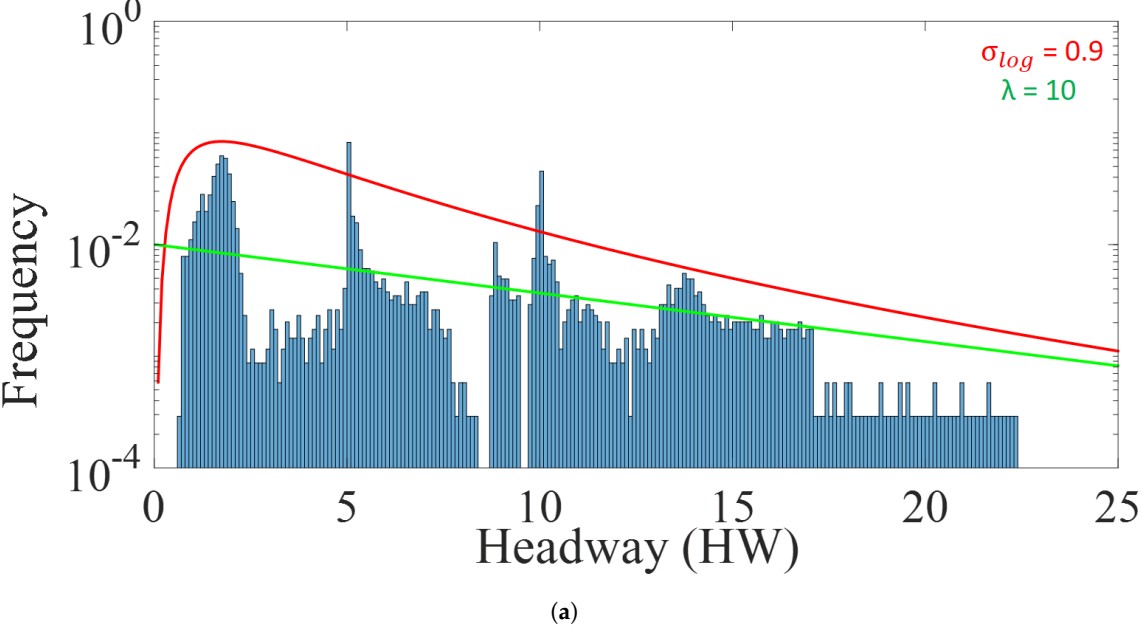

(a)

**Figure 4.** *Cont.*

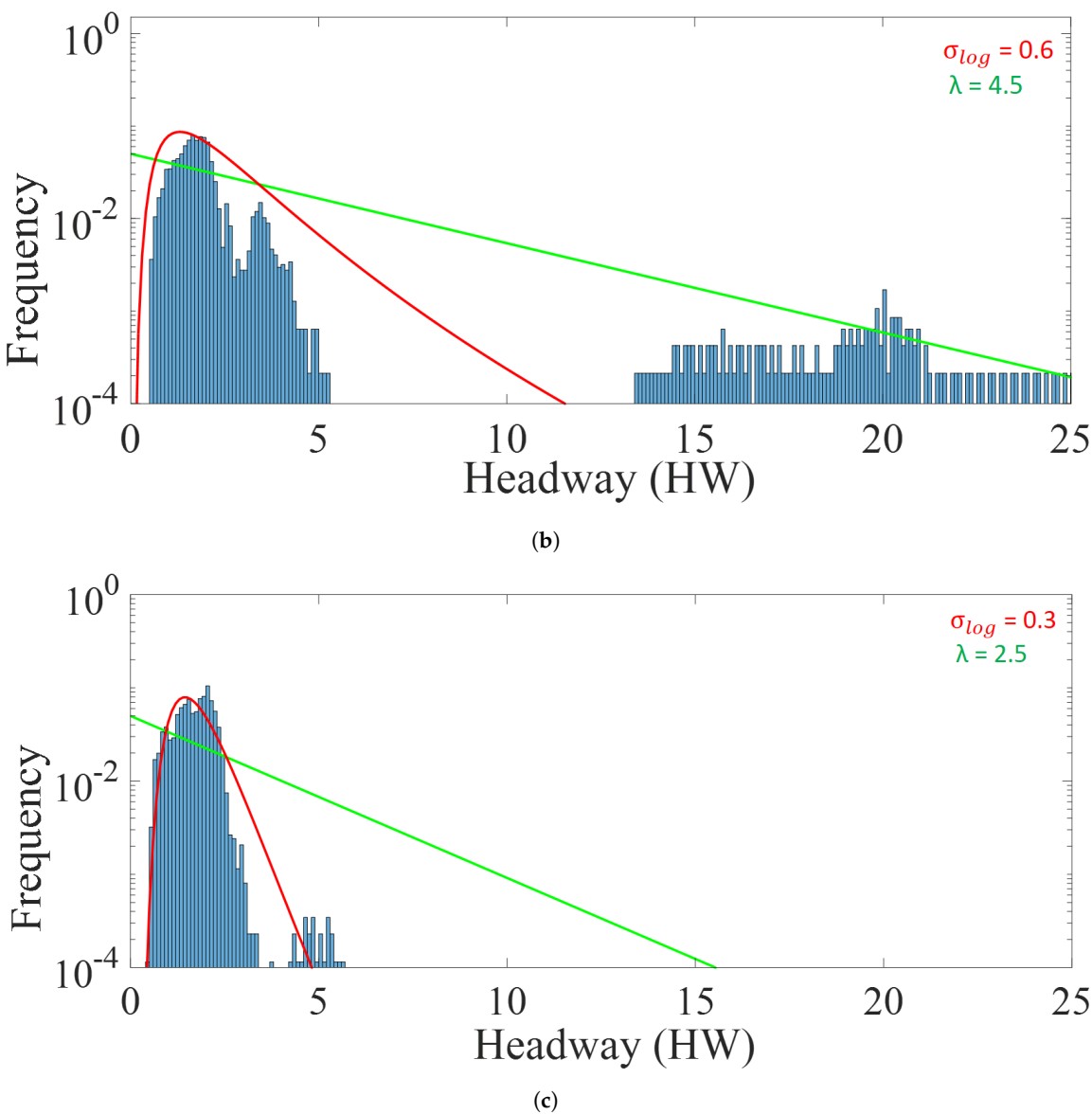

**Figure 4.** The distance headway distribution in different density ranges is plotted: (**a**) $d \in [0, 0.2]$, (**b**) $d \in [0.2, 0.4]$, (**c**) $d \in [0.4, 0.7]$. For all density ranges, the corresponding log normal (red) $((Frequency(HW)) = \frac{1}{HW\sigma_{log}\sqrt{2\pi}}e^{\frac{(D-ln(HW))^2}{2\sigma_{log}^2}}))$ and negative-exponential (green) $(Frequency(HW) = e^{(\frac{-HW}{\lambda})})$ distributions are also shown. Simulation scenarios: $er = 0.02$, $L = 1000$ cells, $\sigma_{sat} = 80$, $v_{min} = 0.15$.

In conclusion, our model captured all the main features of ant traffic presented in [14]: (1) the absence of jamming for a wide range of $d$ values, (2) platoon formations, and (3) a decrease in velocity fluctuations with an increase in the $d$ value, showing that the present model can reasonably replicate empirical data and therefore mimic an AT system under certain limits. Meanwhile, the modeling assumptions from previous physiological studies validated that the agent in our model mimicked ants in real life.

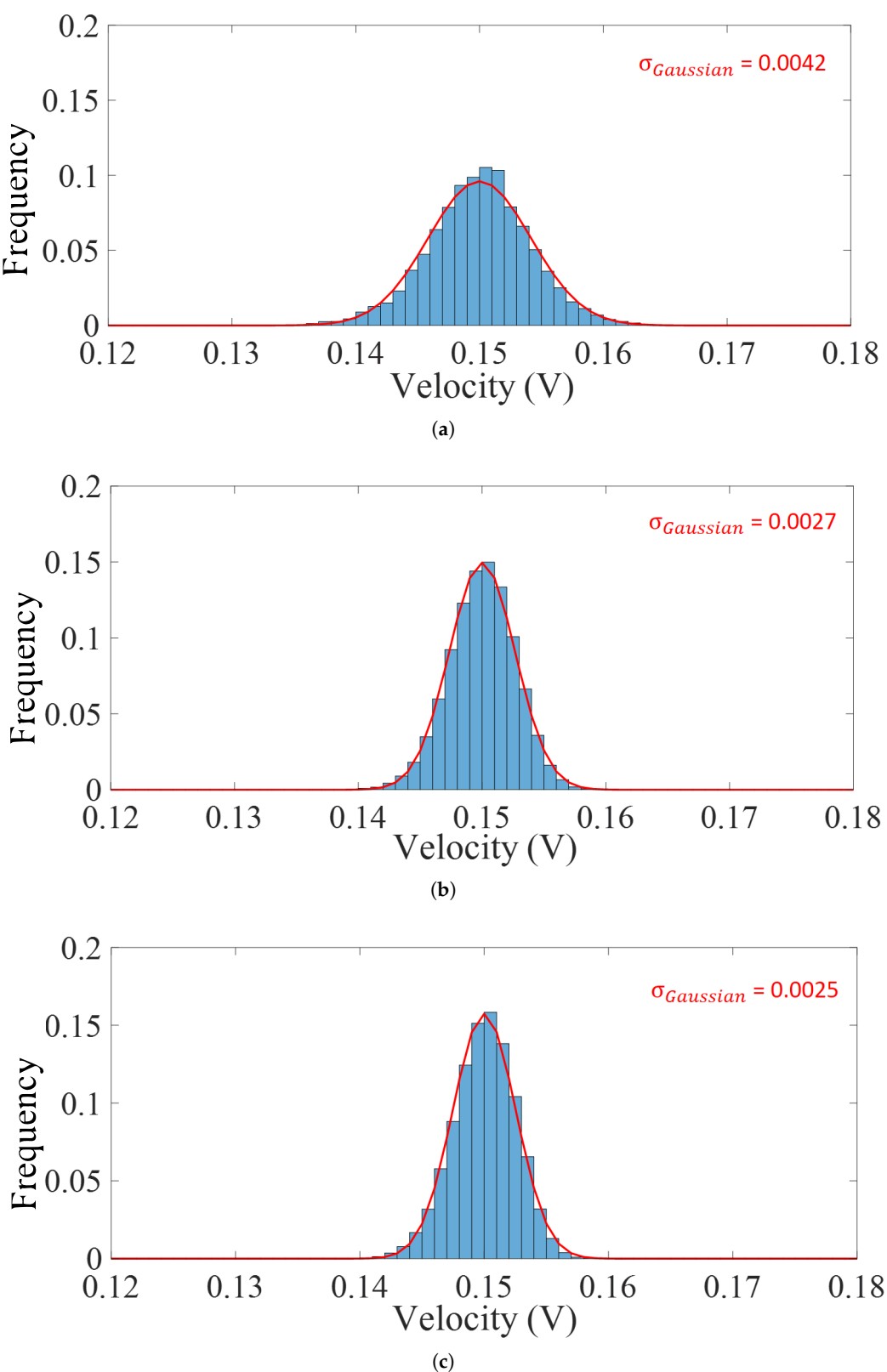

**Figure 5.** The velocity distribution of agents in different densities is plotted: (**a**) $d \in [0, 0.2]$, (**b**) $d \in [0.2, 0.4]$, (**c**) $d \in [0.4, 0.7]$. For all density ranges, the corresponding Gaussian distribution ($Frequency(v)$) $= \frac{1}{\sigma\sqrt{2\pi}} e^{\frac{-(V-v)^2}{2\sigma^2}}$) is also shown. Simulation scenarios: $er = 0.02$, $L = 1000$ cells, $\sigma_{sat} = 80$, $v_{min} = 0.15$.

## 5. Analysis of Jam-Free Ant Traffic

### 5.1. Intra-Platoon Analysis

In the early studies, it was argued that individual ants on a trail have convergent behavior, which leads to the formation of multiple platoons [26]. However, the more recent studies argued that not just individual ants, but also platoons have convergent behavior, and if given sufficient time, all the platoons merge to form a single platoon [5,12,14,27]. It is also argued that ants inside a platoon march together (move with the same $v_j(t)$) [14]. As explained earlier, in the paper, the above two arguments—platoon formation and marching ants—are collectively called the marching platoon hypothesis. In the case of the marching platoon scenario, all the ants in the same platoon should have the same $v_j(t)$. Therefore, to test the hypothesis, we observed the $v_j(t)$ of different ants from the same platoon in an ATM simulation. For this analysis, we chose three ants ($ant_0$, $ant_{50}$, and $ant_{100}$) from the same notional platoon: $ant_0$ was the leader of the platoon, while $ant_{100}$ was the last ant in the platoon. The environment of the selected ants was kept undisturbed, and a simulation was carried out on a heterogeneous trail. In this scenario, $L$ was 1000 units, of which the section from $cell_{400}$–$cell_{800}$ was the high-resistance part ($\Omega = 0.1$). Therefore, the junction between $cell_{399}$ and $cell_{400}$ marked the beginning of a bottleneck, and the junction between $cell_{800}$ and $cell_{801}$ marked its downstream end. Because the value of $\Omega$ for the high-resistance section ($\Omega = 0.1$) was higher than that for the low-resistance part ($\Omega = 0$), we expected the $v_j(t)$ of an ant in the high-resistance part to be lower than that of one in the other part of the trail. Therefore, changes in the velocities of different ants were observed with respect to the position of an ant on the trail.

As shown in Figure 6, the velocity of the leading ant in the platoon decreased suddenly at the bottleneck ($cell_{399}$), whereas those of the other ants in the platoon decreased well before the bottleneck. This decrease in velocity before the bottleneck demonstrated queuing. At the same time, as shown in Figure 6, near the upstream of the bottleneck, ants showed a large velocity fluctuation, where the velocity fluctuated between zero (stop) and high velocity (go) due to the queuing effect. The fluctuation due to the queuing effect indicated stop-and-go motion upstream of the bottleneck. This happened because the flow out of the bottleneck (i.e., the flow from $cell_{399}$) was lower than the flow into it (i.e., the flow towards $cell_{399}$). This led to the formation of a queue, which forced incoming ants to reduce their velocity as they approached the bottleneck. For a given ant in a platoon, the more ants in front there were, the longer the queuing experienced. This means that ants in a platoon (other than the leader) do experience stop-and-go motion. It is also interesting to note that different ants in the platoon showed differing velocity variation: $ant_0$ had a meager velocity variation where the velocity depended only on the trail conditions ($\Omega$), whereas the variations for $ant_{50}$ and $ant_{100}$ were much larger and far from identical. Contrary to previous perceptions, the above dissimilarity of velocity variation indicated that ants in a platoon do not march in sync. This desynchronizing behavior is consistent with the notion that the movements of ants on a trail is based on the pheromone concentration in the next cell, which depends on the flow of ants in the recent past. Therefore, ants cannot have real-time information about other ants in a platoon and so cannot synchronize their actions as a group.

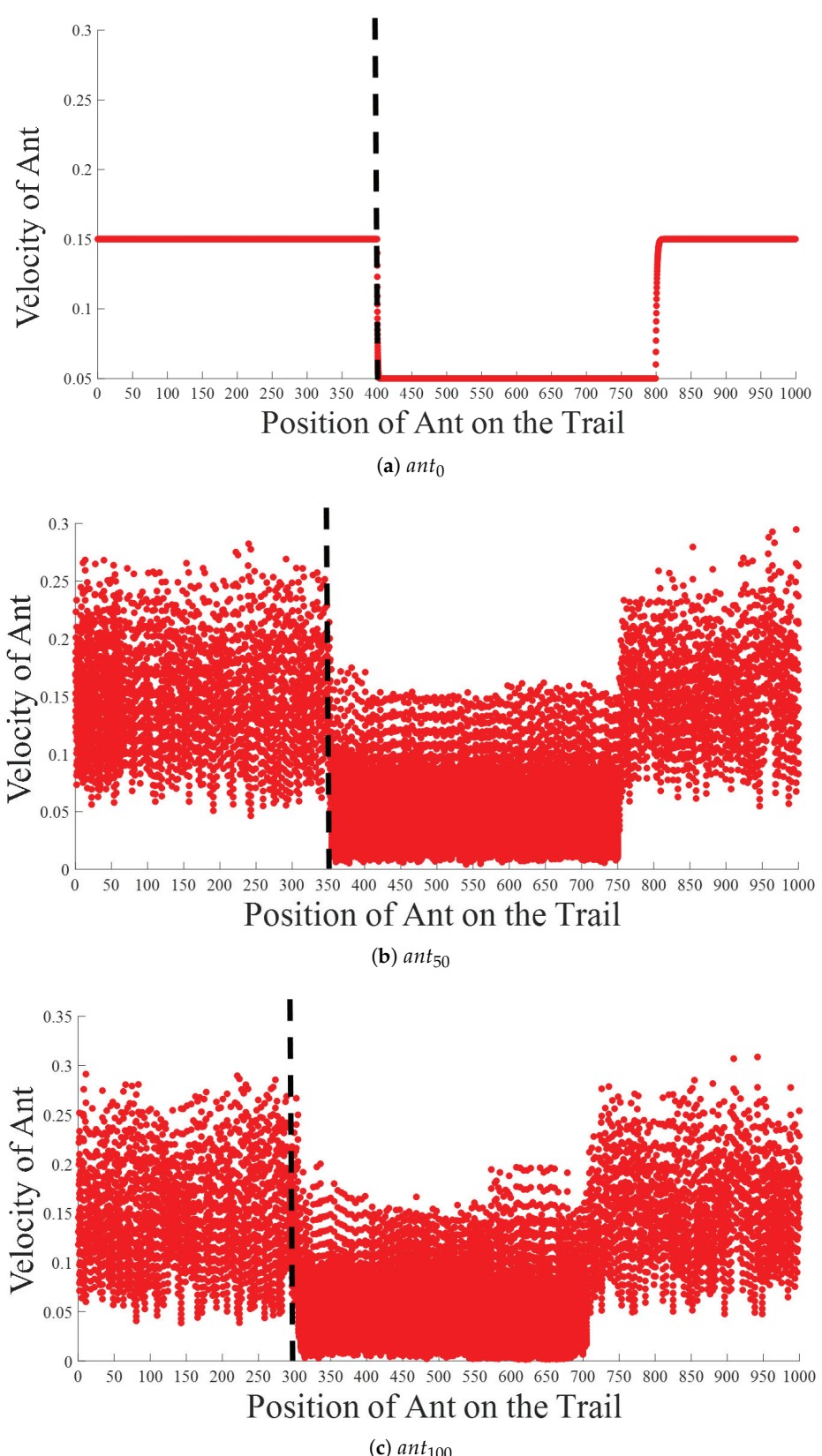

(**a**) $ant_0$

(**b**) $ant_{50}$

(**c**) $ant_{100}$

**Figure 6.** The instantaneous velocities of (**a**) $ant_0$, (**b**) $ant_{50}$, and (**c**) $ant_{100}$ in the intra-platoon analysis plotted against the positions of the same ants. The presented data were extracted from an ATM computer simulation of a heterogeneous trail. Simulation scenarios: $er = 0.02$; high-resistance section $(\Omega = 0.1)$ of trail $= cell_{400}$–$cell_{800}$, $L = 1000$ cells, $\sigma_{sat} = 80$, $v_{min} = 0.15$.

### 5.2. Inter-Platoon Analysis

Contrary to the previous perception, the intra-platoon analysis presented above showed no synchronization between ants in a platoon. Furthermore, evidence of stop-and-go motion was found due to queuing. Therefore, to understand AT systems in more detail, we analyzed inter-platoon relationships. For the inter-platoon analysis, we selected a pair of adjacent platoons that we referred to as the leading platoon and the following platoon, considering the direction of travel. We simulated the heterogeneous scenario described in the intra-platoon analysis. We selected a pair of platoons that were separated by a sufficient distance to ensure that they would not merge into each other in the free-flow phase. We analyzed an intersection of two adjoining platoons, which included (1) an analysis of the last ant in the leading platoon, (2) an analysis of the space in between the two platoons (i.e., the headway of the following platoon), and (3) an analysis of the first ant in the following platoon. We conducted the above-mentioned analysis by observing the data of the last ant in the leading platoon and the first ant in the following platoon.

Prototypical results of the inter-platoon analysis of the free-flow phase are shown in Figure 7, where (a) the velocity–position relationships and (b) the space–time relationships for the last ant in the leading platoon and the first ant in the following platoon are compared. As shown in Figure 7a, although the velocity of the last ant in the leading platoon decreased well before the bottleneck (for the reason given in the intra-platoon analysis above), the velocity of the leader of the following platoon decreased suddenly at the bottleneck. At the same time, concurring with the above observation, the upstream part of the bottleneck in Figure 7b also shows that although the inter-platoon distance decreased, it did not decrease enough to force the leader of the following platoon to reduce its velocity. This means that the velocity of the leader of the following platoon was independent of the velocity of the last ant in the leading platoon, and stop-and-go motion due to queuing in the leading platoon did not affect the following platoon. This phenomenon can be explained by jam-absorbing driving in [30,31]. In jam-absorption driving from [30,31], in response to a trigger of stop-and-go motion (e.g., sudden braking), the following driver creates enough headway to avoid stop-and-go motion. In AT, the inter-platoon distance (referred to as the jam absorption buffer (JAB)) acts similarly to the above-mentioned headway of jam-absorbing driving. The JAB provides enough time for dissipation of the queuing effect due to the leading platoon. Thus, the leader of the following platoon does not get involved in stop-and go-motion due to the leading platoon. Furthermore, in Figure 7, it is interesting to note that at the downstream end of the bottleneck, the velocity of the last ant in the leading platoon rose earlier and was higher compared to the leader of the following platoon. This is where a JAB was created, indicating that a fast-out action created a JAB at the downstream end of a bottleneck, thereby helping to avoid stop-and-go motion at the upstream end. We also observed that for given densities in the free-flow phase, the leaders of all platoons in established traffic flow moved with the same ($v_{avg}$), which led to the same $v_{avg}$ for all platoons. The similar $v_{avg}$ in established traffic flow prohibited the platoon from converging. The above observation about non-convergence is contradictory to previous studies, which argued that platoons in AT have convergence behavior, which leads to the formation of a single platoon [5,12,14,27].

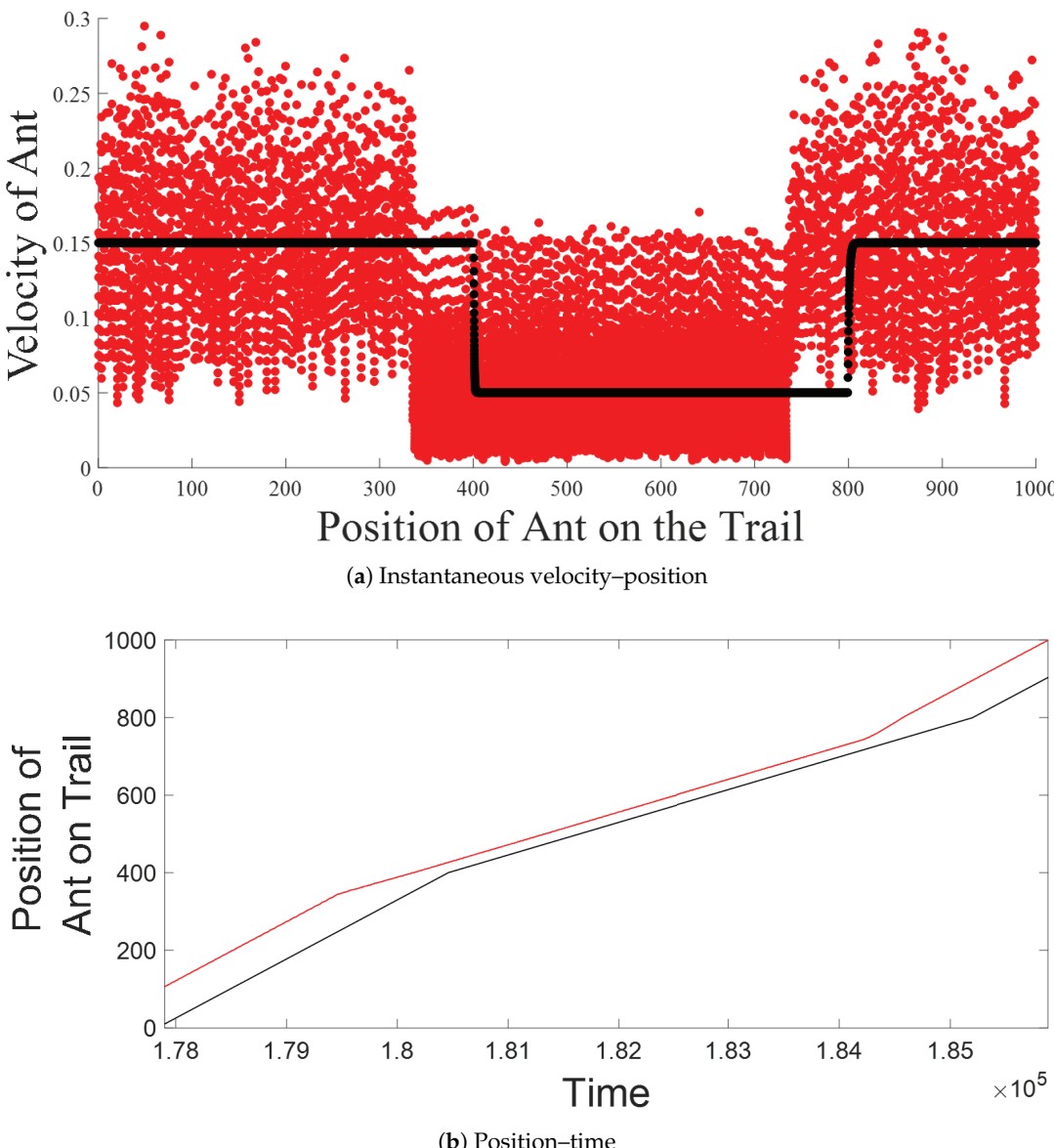

(**a**) Instantaneous velocity–position

(**b**) Position–time

**Figure 7.** (**a**) Instantaneous velocities of ants plotted against the positions of the same ants for prototypical inter-platoon analysis. (**b**) The positions of the ants plotted against time for prototypical inter-platoon analysis. Data were obtained from an ATM computer simulation of a heterogeneous trail. Ants observed: last ant in leading platoon (red); leader of the following platoon (black). Simulation scenarios: $er = 0.02$, high-resistance section ($\Omega = 0.1$) of trail = $cell_{400}$–$cell_{800}$, $L = 1000$ cells, $\sigma_{sat} = 80$, $v_{min} = 0.15$.

## 5.3. Analysis of Platoon Headway and Density

The inter-platoon analysis indicated that an ant colony maintains $f$ on a trail by implementing a jam absorption mechanism, for which the platoon headway (i.e., the JAB) plays an important role. Therefore, in this part of the paper, we present further analysis of the JAB. Similar to the inter-platoon analysis, we selected a pair of adjacent platoons with a sufficient inter-platoon distance (i.e., to avoid merging). The simulation was carried out with a heterogeneous trail, and we took care to ensure that the given JAB was undisturbed (i.e., no new ants were introduced into it). Changes in the JAB were observed, and the number of platoons was counted for each observed density.

The results of analyzing the relationship between JAB and *d* are presented in Figure 8 and Table 3. As shown in Figure 8, in the free-flow phase (up to $d \approx 0.75$), first, a lengthening of the JAB was observed until $d \approx 0.5$. Then, after that, a slight decrease in the JAB length with a rising *d* was observed. It is important to note that the decrease was small when compared against the initial lengthening. At the same time, as shown in Table 1, the number of platoons on the trail decreased.

An increase in *d* led to a larger platoon size, meaning that adjoining platoons that were close to each other came closer and finally merged to form a single platoon, leading to fewer platoons overall. Meanwhile, the platoon mergers provided more space for the remaining platoons to create larger JABs. This JAB lengthening played an important role in managing the *f* on a trail. The platoon size increased with *d*, leading to longer queues. However, the aforementioned JAB lengthening canceled this queue lengthening, thereby maintaining an approximately constant $v_{avg}$. Finally, near a critical density of $d \approx 0.75$, any further increase in *d* led to the collapse of the multi-platoon structure to form a single platoon (infinite cluster) on a circular trail. Because multiple platoons no longer exist, the stop-and-go motion of one ant propagated to the following ant without being absorbed. This was amplified at each ant and traveled backward as a shock wave. Therefore, jams were never absorbed and were amplified further with *d*. Consequently, the $v_{avg}$ decreased with *d*, triggering the jamming of the fundamental diagrams.

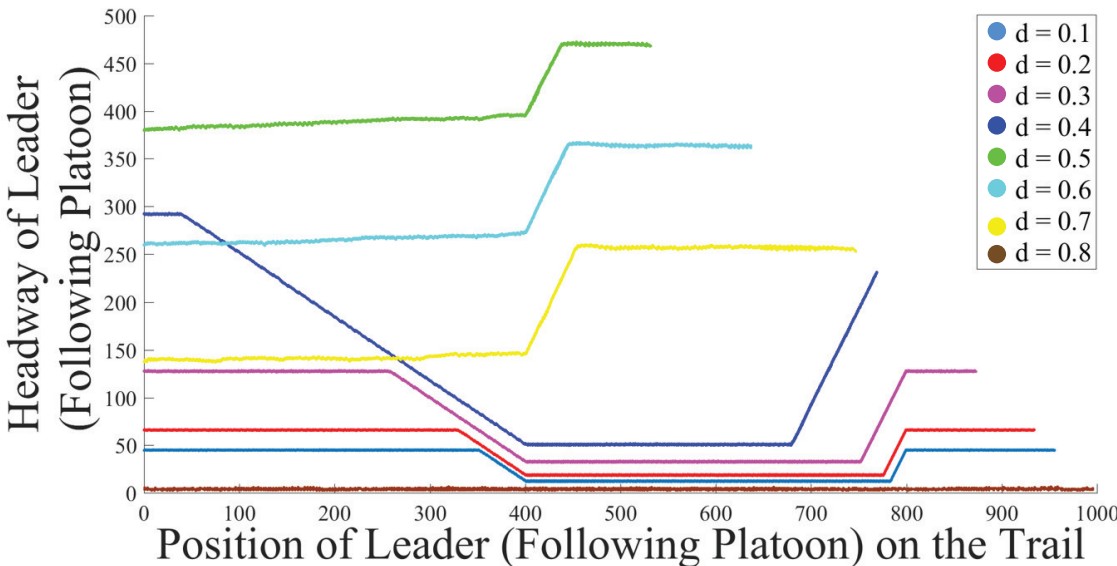

**Figure 8.** Headway (jam absorption buffer) of the leader of the following platoon for different densities (*d*) plotted against the position of the same ant for inter-platoon analysis. Simulation scenarios: $er = 0.02$, high-resistance section ($\Omega = 0.1$) of trail = $cell_{400}$–$cell_{800}$, $L = 1000$ cells, $\sigma_{sat} = 80$, $v_{min} = 0.15$.

**Table 3.** Number of platoons with respect to density.

| Density | Number of Platoons |
|---------|--------------------|
| 0.1 | 28 |
| 0.2 | 21 |
| 0.3 | 14 |
| 0.4 | 8 |
| 0.5 | 3 |
| 0.6 | 3 |
| 0.7 | 2 |
| 0.8 | 1 |

## 6. Concluding Discussion

We presented an agent-based model inspired by ants' chemotaxis behavior on a unidirectional single-lane ant trail to understand the jam-free AT presented in [14]. Our model was an improvement over the model in [9], which was a general model of ant traffic. Our modeling assumptions were based on previous physiological studies, and our model was in basic agreement with the experiment from [14]: (1) the absence of jamming for a wide range of densities, (2) platoon formations, and (3) a decrease in velocity fluctuations with an increase in density. We also presented a computer-simulated analysis of different platoon scenarios to understand the platoon-related phenomena. The primary objectives of these analyses were to verify the marching-platoon hypothesis and to understand the mechanism of jam-free AT.

Similar to previous studies, our study also mimicked the platooning and constant average velocity of AT for a wide range of densities ($d = 0$–$\approx 0.75$). However, contrary to previous studies, the intra-platoon analysis presented in this paper showed that ants in a platoon (other than the leader) might be experiencing stop-and-go motion. It was also interesting to note that different agents in a platoon had different variations in their velocities. This difference means that contrary to the previous perception, ants in a platoon might not be marching synchronously.

Meanwhile, the inter-platoon analysis of ATM presented in this paper shows that ants' chemotaxis behavior leads to a jam absorption mechanism, which helps to avoid jamming. In the jam absorption mechanism of ATM, the inter-platoon distance (i.e., the JAB) provided enough time for the system to dissipate the queuing effect due to leading platoons. Thus, the following platoons did not get involved in stop-and-go motion due to those leading platoons.

Further inter-platoon analysis of ATM from a JAB-density perspective showed that the JAB in the free-flow phase increased with density, thereby canceling out the increase in queues due to density. Finally, we also observed that near a critical density of $d \approx 0.75$, the multi-platoon structure collapsed to form a single platoon (infinite cluster). This caused the stop-and-go motion of one ant to propagate to the following ant without being absorbed. Consequently, the average velocity decreased with density, triggering the jamming phase of the fundamental diagrams.

Our model provided a detailed understanding of the dynamics of ants on an ant trail. The new findings of our study need to be verified and validated by using different methods. Nevertheless, our results may have significant implications for swarm intelligence and intelligent transportation systems. In future work, we intend to understand the JAB by analyzing different related phenomena.

**Author Contributions:** Conceptualization, P.K. and H.N.; modeling, simulation, and validation, P.K.; writing, original draft preparation, P.K.; supervision and critical review, H.N.

**Funding:** This research received no external funding.

**Conflicts of Interest:** The authors declare no conflict of interest.

## Appendix A. Variables in the Ant Trail Model

### Appendix A.1. Minimum Velocity in ATM ($v_{min}$)

As shown in Figure A1, with an increase in $v_{min}$, the $v_{avg}$ increases. At the same time, the velocity fluctuation and critical density decrease. Inside a platoon in the ATM, the upper limit of agent velocity is governed by the velocity of the platoon leader. In established traffic on ATM (with low *er*), the leader of a platoon does not experience pheromones in the free-flow phase; thus, the leaders travel with $v_{min}$. Therefore, with an increase in $v_{min}$, the velocity of all ants in the platoon increased, which led to a higher $v_{avg}$. At the same time, the velocity range between the saturation level of velocity and $v_{min}$ decreased, leading to a decrease in velocity fluctuation. It is also interesting to note that with an increasing $v_{min}$, the critical density decreased and the fundamental diagrams' resemblance with TASEP increased. Eventually, at a high $v_{min}$ ($v_{min} > 0.5$), the ATM behaved similarly to the TASEP model.

With the decrease in the velocity range, more ants in the simulation moved with a similar velocity to the saturation level velocity, leading to a similar simulation to TASEP with a high hopping probability.

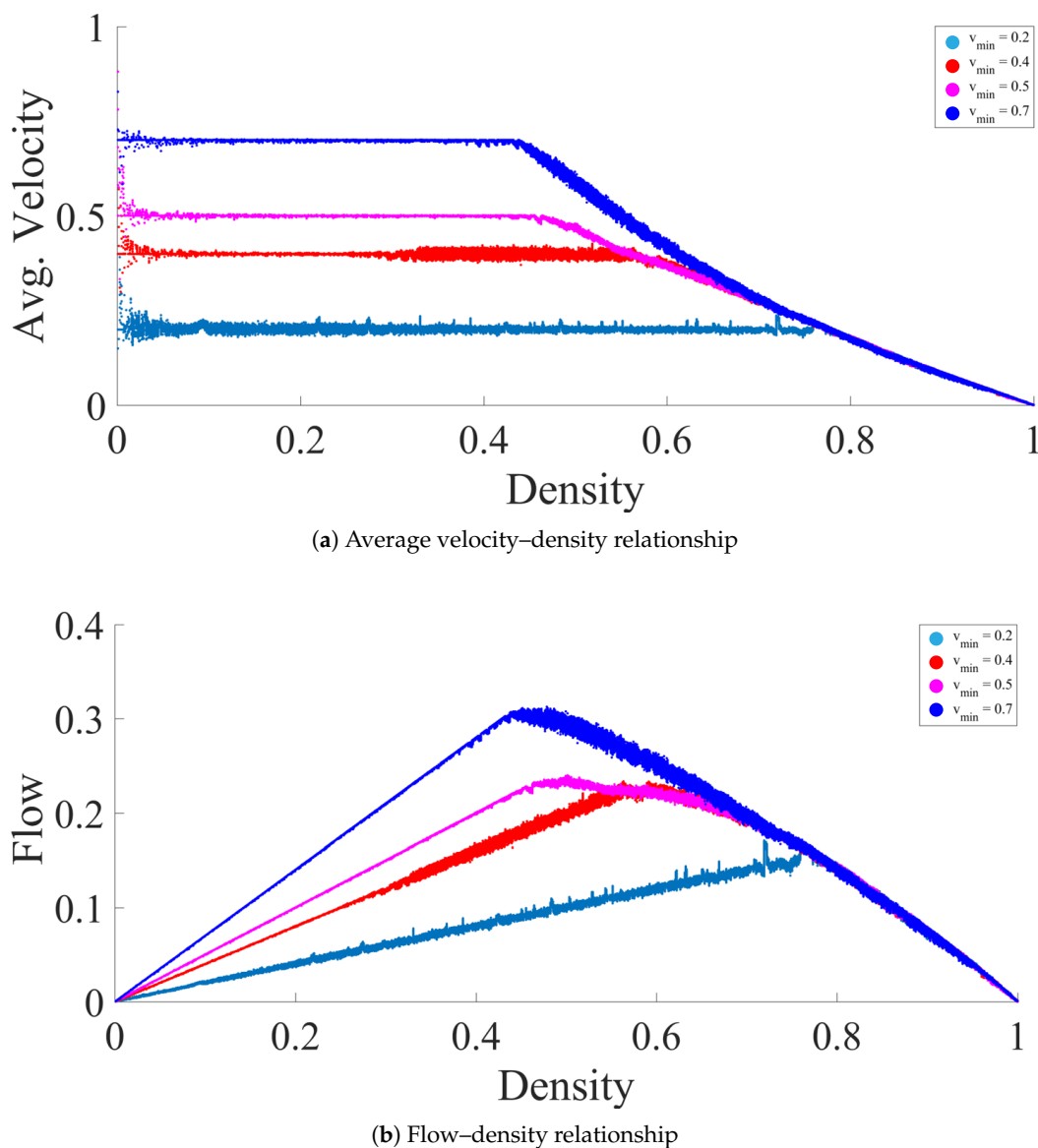

(**a**) Average velocity–density relationship

(**b**) Flow–density relationship

**Figure A1.** (**a**) Average velocity and (**b**) flow of agents plotted against their densities for different $v_{min}$ values (indicated in the legend). Parameters other than $v_{min}$ were kept constant: $L = 1000$ cells, $\sigma_{sat} = 80$, $er = 0.02$.

*Appendix A.2. Pheromone Saturation Level ($\sigma_{sat}$)*

The maximum possible velocity of the agents in the ATM depends on $\sigma_{sat}$. Thus, from a system perspective, the capacity flow and capacity flow density depend on $\sigma_{sat}$. As shown in Figure A2, at $\sigma_{sat} = 1$, ATM had a similar capacity flow and capacity flow density as TASEP with a low hopping probability (Figure 2). The above similarity happened because at $\sigma_{sat} = 1$, even a small amount of evaporation reduced the pheromone concentration to undetectable levels ($\sigma < 1$), leading to the condition where most of the agents traveled without pheromones (constant low velocity). In this scenario, ATM behaved similarly to TASEP with a low hopping probability. After $\sigma_{sat} = 1$, initially, with the rise in $\sigma_{sat}$ for the range $1 \leq \sigma_{sat} \leq 5$, a sharp rise in capacity flow, as well as capacity flow density was observed; thereafter, for a wide range of $\sigma_{sat}$ ($5 \leq \sigma_{sat} \leq 80$), both the capacity flow and capacity flow density remained constant. Based on the above observations, we can state that, for a

considerable range of $\sigma_{sat}$ ($5 \leq \sigma_{sat} \leq 80$), ATM simulations were independent of $\sigma_{sat}$. We intend to analyze the above behavior of ATM concerning $\sigma_{sat}$ in future work; however, for the purpose of this paper, the above behavior analysis was considered as out of scope.

Then, afterwards, at a very high $\sigma_{sat}$ ($\sigma_{sat} > 80$), the capacity flow and capacity flow density started decreasing, and eventually ($\sigma_{sat} > 4000$), the ATM again behaved similarly to TASEP with a low hopping probability. This happened because at very high $\sigma_{sat}$ values, due to the effect of pheromone evaporation, the pheromone never accumulated and remained at a comparatively meager value, which led to a low velocity. In this scenario, the pheromone became ineffective; hence, most of the agents behaved similarly to agents in TASEP with a low hopping probability.

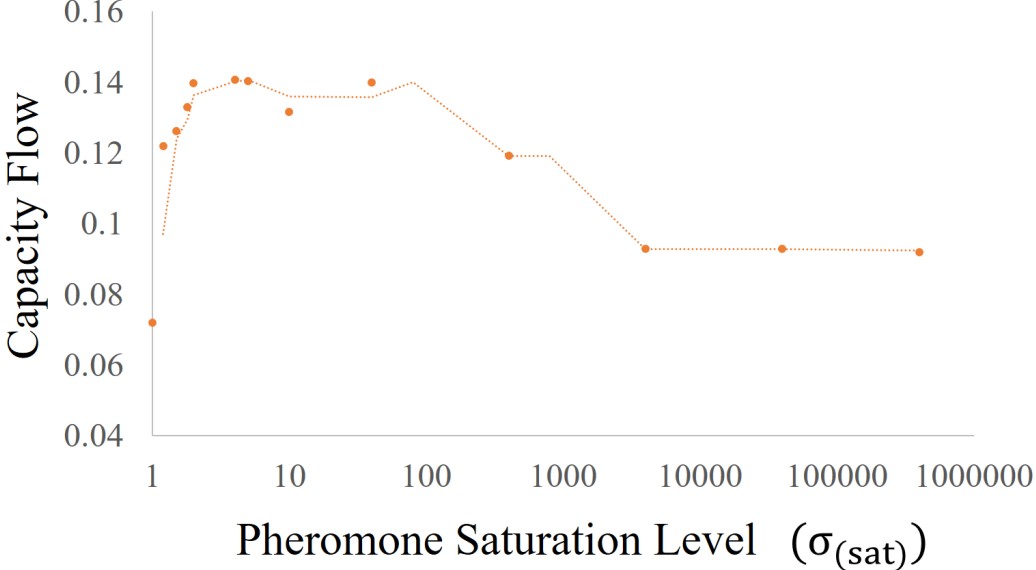

(**a**) Capacity flow–pheromone saturation level ($\sigma_{sat}$)

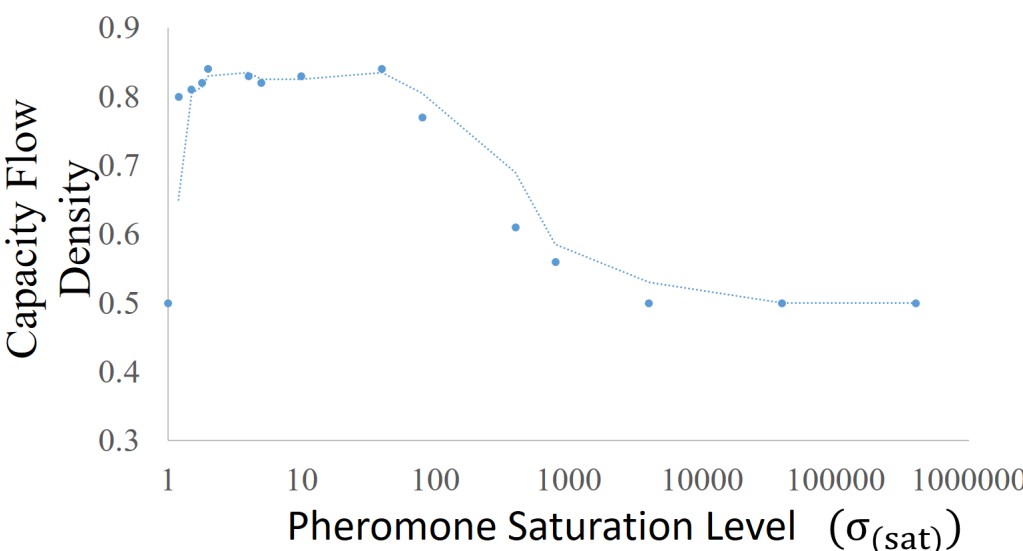

(**b**) Capacity flow density–pheromone saturation level ($\sigma_{sat}$)

**Figure A2.** The (**a**) capacity flow and (**b**) capacity flow density in the simulation are plotted against $\sigma_{sat}$. Parameters other than $\sigma_{sat}$ were kept constant: $L$ = 1000 cells, $v_{min}$ = 0.15, $er$ = 0.02.

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
