# Peer review of "Congestion-Free Ant Traffic: Jam Absorption Mechanism in Multiple Platoons"

_applsci, doi:10.3390/app9142918_

Round 1
Reviewer 1 Report
This manuscript presents a model of ant traffic on a unidirectional single-lane ant trail to understand the jam-free traffic. The authors provide a significant improvement from the previous similar model and validate the results with the results of previous empirical study. The topic is very interesting, and the manuscript is mostly well-structured, but it is a bit mixing between the method and the results. There are many flaws that demand to be addressed before acceptance. My detailed comments are as follows.
Major comments:
1. I am doubtful to say that this research uses agent-based modelling (ABM) (Line 1) as the author said that this is the improvisation model from Chowdhury et al. [1] (Line 51). Chowdhury et al. used Cellular Automata (CA) on their model which is clearly used in this paper. Ant-trail model (ATM) used in this model is similar to CA-based ant-trail model from Chowdhury et al. [1] where, as stated by the authors, there is no difference on the modelling approach between this paper and one by Chowdhury et al. except the improvisation on several aspects. Yes, CA and ABM share similarities that might be confusing, but there are differences among them [2,3].
2. If this is an ABM, the authors omit, or might be unclear, any description of several aspects about the environment, how the agent interacts with others and with the environment (interaction), and the behaviour rule of the agent. Moreover, the description of how the agent's generation and initialisation are missing from this paper.
3. The simplicity of a model is not a strong reason for a model to be improved [4] (see Line 46), because the purpose of a model is to simplify a complex real world. The authors should have a strong stand on why this improvement is important.
4. How can the cells and trails be called as agents? (Line 87) I think these supposed to be environment.
5. The purpose of this paper is to validate the model and test marching platoon hypothesis, but there is no clear explanation of the method of validation (Line 175).
6. I couldn’t find the graphic representing the comparison between the result from this model with one from John et al. [5] (see Line 280).
7. The author said that there is evidence of stop-and-go motion due to queuing (Line 339), but this argument was not supported with data.
8. I was a bit confusing when reading the conclusion. Is it actually the discussion section? Also, please avoid redundant writing in the conclusion.
Minor comments:
Never the less (line 439) should be Nevertheless.
Line 68: arises?
Line 197: veriable should be variable.
References
1. Chowdhury, D.; Guttal, V.; Nishinari, K.; Schadschneider, A. A cellular-automata model of flow in ant trails: non-monotonic variation of speed with density. J. Phys. Math. Gen. 2002, 35, L573.
2. Crooks, A.; Malleson, N.; Manley, E.; Heppenstall, A. Agent-Based Modelling and Geographical Information Systems: A Practical Primer; SAGE, 2019; ISBN 978-1-5264-5416-4.
3. Crooks, A.T.; Heppenstall, A.J. Introduction to agent-based modelling. In Agent-based models of geographical systems; Springer, 2012; pp. 85–105.
4. Sun, Z.; Lorscheid, I.; Millington, J.D.; Lauf, S.; Magliocca, N.R.; Groeneveld, J.; Balbi, S.; Nolzen, H.; Müller, B.; Schulze, J.; et al. Simple or complicated agent-based models? A complicated issue. Environ. Model. Softw. 2016, 86, 56–67.
5. John, A.; Schadschneider, A.; Chowdhury, D.; Nishinari, K. Trafficlike collective movement of ants on trails: Absence of a jammed phase. Phys. Rev. Lett. 2009, 102, 108001.
Reviewer 2 Report
A good paper and well presented. I enjoyed reading it.
Reviewer 3 Report
The report is attached as a .pdf file.

Reviewer 4 Report
The paper proposes an agent-based model of ant traffic on a unidirectional single-lane ant trail to understand the jam-free traffic of an ant colony. Results show that ants on a trail do not march synchronously and do experience stop-and-go motion. Moreover, the study indicates that the ants’ chemotaxis behavior leads to peculiar jam absorption mechanism, which helps maintain free flow on a trail and avoids jamming.
As a general comment, I think the paper makes an interesting contribution to the literature. The model and results section are clearly explained. At the same time, I think the paper needs some minor improvements before to be published in this journal.
Broad comments
1) The agent-based models are a very useful methodology in study complexity in many fields. I would suggest the author to add at least a paragraph in the introduction where this topic should be mentioned. I suggest also to add the following citations:
a. Jennings, Nicholas R. "On agent-based software engineering." Artificial intelligence 117.2 (2000): 277-296.
b. Ponta, Linda, and Silvano Cincotti. "Traders’ networks of interactions and structural properties of financial markets: an agent-based approach." Complexity 2018 (2018).
c. Batty, Michael. Cities and complexity: understanding cities with cellular automata, agent-based models, and fractals. The MIT press, 2007.
2) In section 2 I would suggest the authors to add a table summarizing the main variables of the model.
3) In order to let the paper more readable, I would suggest the authors to add a table with the main values used in the simulation scenarios.
Specific comments
Before starting the list of references a title “References” should be added.
Round 2
Reviewer 1 Report
The author has made great improvements in accordance with the previous review report. However, I recommend to ask the authors to make their ABM source code available online either along with this article or separated on the other repository for examples OpenABM, GitHub, etc.